# DNA Methylation Profiling in Rare Sellar Tumors

**DOI:** 10.3390/biomedicines10092225

**Published:** 2022-09-08

**Authors:** Kyla Wright, Kristyn Galbraith, Matija Snuderl, Nidhi Agrawal

**Affiliations:** 1NYU Grossman School of Medicine, New York, NY 10012, USA; 2NYU Langone Medical Center, New York, NY 10012, USA

**Keywords:** pituitary tumor, esthesioneuroblastoma, sella turcica, DNA methylation

## Abstract

The histologic diagnosis of sellar masses can be challenging, particularly in rare neoplasms and tumors without definitive biomarkers. Moreover, there is significant inter-observer variability in the histopathological diagnosis of many tumors of the CNS, and some rare tumors risk being misclassified. DNA methylation has recently emerged as a useful diagnostic tool. To illustrate the clinical utility of machine-learning-based DNA methylation classifiers, we report a rare case of primary sellar esthesioneuroblastoma histologically mimicking a non-functioning pituitary adenoma. The patient had multiple recurrences, and the resected specimens had unusual histopathology. A portion of the resected sellar lesion was profiled using clinically validated whole-genome DNA methylation and classification. DNA was extracted from the tissue, hybridized on DNA methylation chips, and analyzed using a clinically validated classifier. DNA methylation profiling of the lesion showed that the tumor classified best with the esthesioneuroblastoma reference cohort. This case highlights the difficulty in diagnosing atypical sellar lesions by standard histopathological methods. However, when phenotypic analyses were nonconclusive, DNA methylation profiling resulted in a change in diagnosis. We discuss the growing role of DNA methylation profiling in the classification and diagnosis of CNS tumors, finding that utilization of DNA methylation studies in cases of atypical presentation or diagnostic uncertainty may improve diagnostic accuracy with therapeutic and prognostic implications.

## 1. Introduction

The differential for a lesion of the sellar region encompasses a range of neoplastic and non-neoplastic entities arising from the pituitary gland itself, sinonasal epithelium, or adjacent anatomical structures. Pituitary adenomas are the most common sellar lesions. Non-pituitary masses include craniopharyngiomas, meningiomas, chordomas, chondrosarcomas, schwannomas, and plasmacytomas, as well as non-neoplastic lesions such as cysts and inflammatory lesions. The diagnosis can be challenging as the clinical and radiological features of these entities are not specific and can overlap significantly [1]. While histopathological diagnosis is key, it is rarely nonconclusive, necessitating the need for further testing.

DNA methylation profiling has emerged as a useful tool in the classification, sub-grouping, and prognostication of many central nervous system (CNS) lesions [2]. It has been reported to improve diagnostic precision compared with standard methods [3]. Moreover, medulloblastomas, ependymomas, and meningiomas have well-defined subgroups based on DNA methylation profiling that can predict tumor recurrence and prognosis [2,4]. Further, DNA methylation profiling of extracellular vesicles or cell-free DNA has been proposed as a biomarker to identify tumor subgroups via liquid biopsy in glioma and glioblastoma [5,6]. Thus, the role of DNA methylation profiling in the diagnosis and stratification of CNS tumors is growing and continues to be a target of study, with aims to improve both diagnostic and prognostic accuracy.

We report a case of primary sellar esthesioneuroblastoma mimicking a non-functioning pituitary adenoma. The patient’s presentation was atypical, with multiple recurrences and inconsistent pathological diagnoses. However, DNA methylation profiling showed that the tumor did not cluster with pituitary adenoma but with the esthesioneuroblastoma reference cohort. Here, we discuss the clinical and pathological findings of this case as well as highlight the diagnostic utility of DNA methylation profiling for atypical sellar lesions. 

## 2. Case Report

### 2.1. Clinical Presentation

The patient, a 58-year-old female, initially presented with blurry vision for 1.5 years (Table 1). Visual field testing revealed bilateral temporal hemianopsia and subsequent imaging revealed an enhancing sellar and suprasellar mass measuring 2.0 cm × 1.8 cm compressing the optic chiasm. She underwent transsphenoidal resection. The conundrum of her pathological diagnosis will be discussed below. Post-operatively, her vision continued to decline, with an increase in the tumor’s suprasellar component (Figure 1A). She underwent a second resection with successful decompression of the optic apparatus. However, there was residual disease in the cavernous sinus (Figure 1B), for which she subsequently underwent gamma knife radiation. The patient’s post-operative course was complicated by hypopituitarism and the patient was started on replacement steroids and thyroid replacement. Growth hormone replacement was deferred given the risk of tumor growth and patient preference. The patient’s residual tumor remained stable until recent imaging showed progression of the sellar mass (Figure 1C,D).

### 2.2. Phenotypic Analyses

The pathology report from the patient’s first resection found the tumor to have neurocytic differentiation with abundant neuropil and fibrillarity highlighted by neurofilament and synaptophysin staining, with fragments of adenohypophyseal tissue that stained positive for reticulin. The initial pathological diagnosis was ganglioneurocytoma. 

Conversely, pathological examination of the specimen from the patient’s repeat resection suggested a diagnosis of pituitary adenoma with neuronal choristoma (PANCH). PANCH is a descriptive term for a rare tumor showing features of both pituitary adenoma and gangliocytoma and has been hypothesized to arise from ganglionic transdifferentiation of a pituitary adenoma [7] or from progenitor cells with both endocrine and neuronal properties [8]. Adenomatous pituitary and ganglion cells within a substrate of fibrillary neuropil is characteristic [7]. Distinguishing PANCH tumors from pituitary adenomas, the ganglionic tumor cells stain positive for neurofilament (NF) and in anti-NeuN assays [7].

### 2.3. Genomic Analyses

The uncertainty regarding the pathological diagnoses of the patient’s lesion prompted methylation profiling of the neoplasm. The tumor was profiled using clinically validated whole-genome DNA methylation and classification using the brain tumor classifier as previously published [3,9]. In brief, DNA was extracted from the formalin fixed paraffin embedded tissue using Maxwell Promega and hybridized on Illumina EPIC DNA methylation chips. The tissue sample used for analysis was a portion of the surgically resected sellar lesion. DNA was analyzed using a Random Forest classifier. DNA methylation data are available in GEO: GSE211634. 

The classifier used allows for comparison of a diagnostic case with over 2800 reference cases. The output is a classification score between 0 and 1 that indicates the resemblance to one of the included CNS tumor classes. In theory, all class prediction scores for a given diagnostic case should add up to 1. A higher classification score indicates that the diagnostic case resembles a given tumor class. The analysis found our case’s methylation profile to be most consistent with esthesioneuroblastoma with a calibrated score of 0.758. Visual representation of the specimen’s methylation profile in relation to the reference cohort via T-distributed Stochastic Neighbor Embedding (TSNE) clustering generated by the classifier is shown in Figure 2. 

The DNA methylation classifier contains two epigenetically distinct molecular subclasses of esthesioneuroblastoma: subclass A and B. The patient’s specimen was most consistent with esthesioneuroblastoma, subgroup A with a confidence score of 0.425 [9]. Both subclasses are comprised exclusively of tumors with a histological diagnosis of esthesioneuroblastoma occurring in the front-basal region and differ based on chromosomal gains or losses. Nevertheless, the clinical significance of these subclasses is currently unknown (https://www.molecularneuropathology.org/mnp, accessed on 28 November 2021). 

## 3. Discussion

Whole-genome DNA methylation profiling has emerged as a novel molecular approach for the characterization of tumors, improving diagnostic accuracy and reducing inter-observer variability in brain [3] and sinonasal tumors [10]. By comparing a tumor’s DNA methylation profile to a reference set of tumors using machine-learning classifiers, a tumor may be classified with greater diagnostic accuracy compared to standard histopathologic methods [3,11]. Methylation profiles can be uploaded online and instantly analyzed to obtain molecular classification (www.molecularneuropathology.org, accessed on 28 November 2021). In the initial study, the use of the classifier resulted in 12% of cases receiving a change in diagnosis [3]. As tumors of the CNS are further characterized by DNA methylation profile, advocates are calling for implementation of methylation profiling into standard pituitary tumor workups and the next world health organization (WHO) classification of CNS tumor [3,12]. 

Esthesioneuroblastomas are rare malignant neoplasms of the nasal cavity derived from olfactory neuroepithelium [13]. Even rarer is the primary intracranial development of esthesioneuroblastoma outside of the region in which olfactory epithelium exists, including the sellar and parasellar regions [14]. Primary sellar esthesioneuroblastomas mimic other tumors of the sellar region, presenting with endocrine dysfunction, bitemporal hemianopia, and non-specific imaging characteristics [15]. Diagnosis of esthesioneuroblastoma is typically made on histopathology, with well-differentiated tumors showing homogenous small cells with uniform round nuclei, rosette, or pseudo-rosette formation, and an eosinophilic, fibrillary intercellular background [15]. However, undifferentiated tumors are difficult to differentiate from other small-cell neoplasms. Due to lack of definitive markers, esthesioneuroblastomas are frequently misdiagnosed as benign tumors, impacting management and resulting in recurrences [16]. 

The DNA methylation profiles of esthesioneuroblastomas have been analyzed in two prior studies [17,18]. Capper et al. compared the DNA methylation profiles of esthesioneuroblastoma to other tumors of the sinonasal tract, formulating four subtypes of esthesioneuroblastoma based on the expression of cytokeratin and chromogranin A, mutations in IDH2, and DNA methylation patterns [17]. Classe et al. similarly performed an integrated analysis of primary esthesioneuroblastoma samples and proposed two molecular subtypes: the more immature, basal subtype with an IDH2 R172 mutant-enriched subgroup with CpG island methylator phenotype, and the more mature, neural subtype, with genome-wide reprogramming with loss of DNA methylation at enhancers of axonal guidance genes [18]. 

Notably, the DNA methylation classifier results presented in this case were considered indeterminate (confidence score 0.758), whereas a confidence score ≥0.9 is considered positive. This could be due to low tumor cell content, as DNA methylation requires a relatively high tumor cell content and performs best with >70% of tumor cells in the sample [9]. Another possible explanation is that the classifier was trained on a cohort of 39 classic esthesioneuroblastomas localized to the nasal cavity (www.molecularneuropathology.org, accessed on 28 November 2021). Because DNA methylation also reflects the site of origin, a primary sellar esthesioneuroblastoma may exhibit differences in DNA methylation that are yet to be accounted for in the classifier due to their rarity. Cases such as this add to a growing body of DNA methylation profiles shared by the community through www.molecularneuropathology.org (accessed on 28 November 2021), contributing to further refinement of the classifier. 

The case presented here demonstrates a practical application of the DNA methylation-based classifier for CNS tumors developed by Capper et al. The tumor was shown not to be a pituitary adenoma and was classified best as esthesioneuroblastoma instead, resulting in a change in diagnosis. Distinguishing primary sellar esthesioneuroblastomas from benign tumors of the pituitary is important for guiding therapy. Initial therapy for most pituitary adenomas is transsphenoidal resection without adjuvant therapy [19]. On the other hand, in the treatment of esthesioneuroblastoma, surgery with adjuvant radiotherapy results in better recurrence free rates (60–100% versus 14–56%) [20] and 5-year survival (77% versus 61%) [21] compared to surgery alone. Moreover, survival is improved in advanced esthesioneuroblastomas with neoadjuvant or adjuvant chemotherapy and postoperative radiotherapy [22]. 

## 4. Conclusions

Studies have reported inter-observer variability in the histopathological diagnosis of many tumors of the CNS [3,23]. Misdiagnosis is particularly frequent in rare neoplasms or cases without definitive histologic or immunohistochemical markers such as primary sellar esthesioneuroblastomas. This case highlights the difficulty in diagnosing poorly differentiated, atypical sellar lesions by standard histopathological methods. However, when standard histopathologic analyses were nonconclusive, DNA methylation profiling resulted in a change in diagnosis. Thus, utilization of DNA methylation studies in cases of atypical presentation or diagnostic uncertainty may improve diagnostic accuracy with therapeutic and prognostic implications.

## Figures and Tables

**Figure 1 biomedicines-10-02225-f001:**
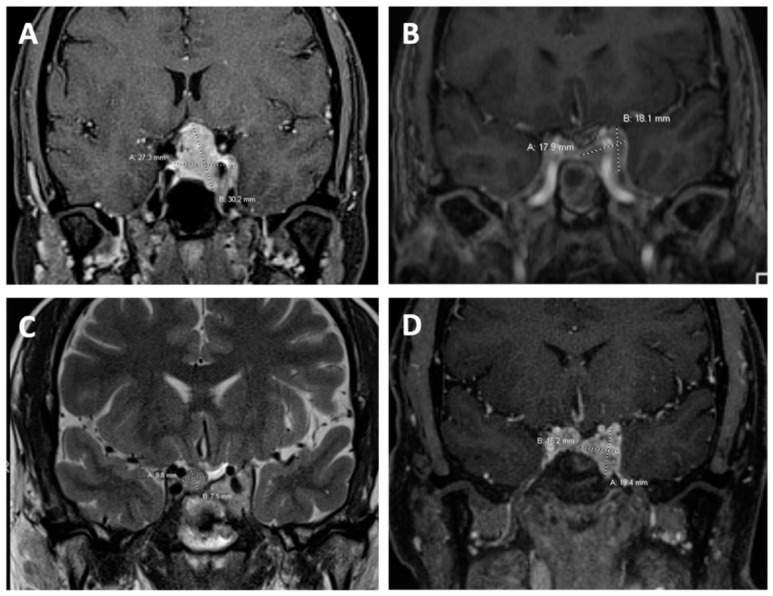
Coronal MR images. (**A**) T1 MR image with 15 cc gadopentetate dimeglumine intravenous contrast taken two years following initial resection showing an enhancing sellar and suprasellar mass (2.2 cm × 2.0 cm × 1.8 cm) that increased in size as per prior report from an outside institution. (**B**) T1 MR image with 4 mL gadobutrol intravenous contrast post repeat transsphenoidal resection of the recurrent mass. There is a stable appearance of the residual lesion (1.6 cm × 1.9 cm). The tumor remained stable until the patients most recent MRI (**C**,**D**). (**C**) T2 weighted MR image and (**D**) T1 MR image with 4.5 mL gadobutrol contrast showing progression of residual tumor with new right sellar and suprasellar mass (1.0 cm × 0.8 cm).

**Figure 2 biomedicines-10-02225-f002:**
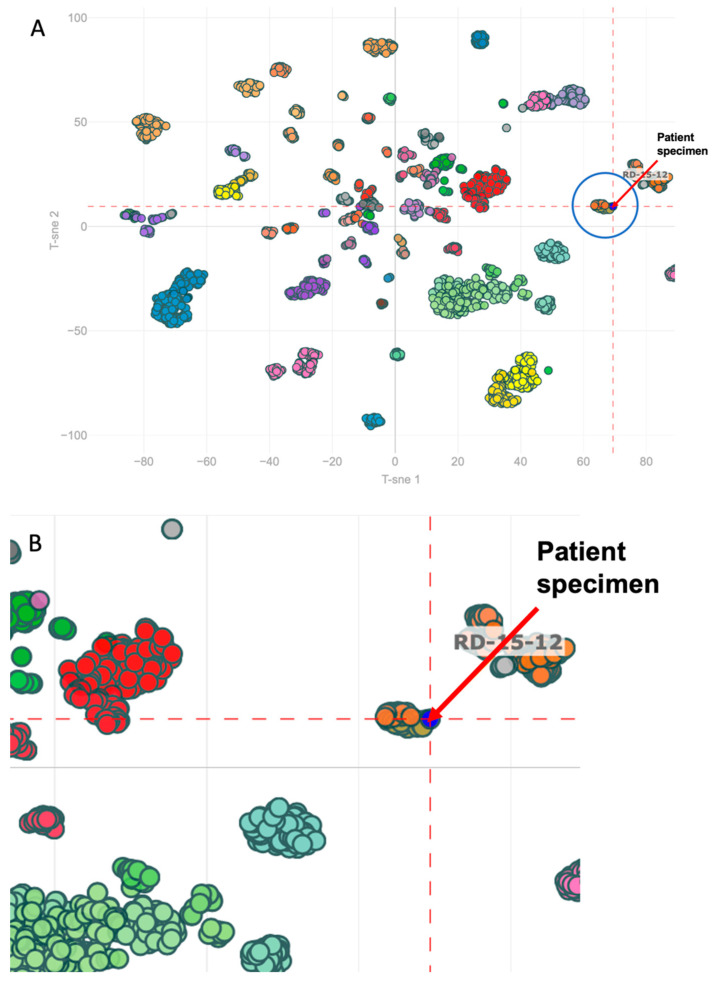
(**A**) TSNE clustering of reference cohort methylation profiles in relation to the patient’s specimen. TSNE clustering is a method for visualizing high-dimensional data in a two-dimensional map. Each point represents a specimen’s methylation profile. The blue point indicated by the red arrow corresponds to our patient’s specimen. All other points correspond to cases in the classifier’s reference cohort. Each tumor class is assigned a color-code. The blue circle encompasses the cluster of reference esthesioneuroblastomas. (**B**) Zoomed-in TSNE clustering to better illustrate ENB subgroups A and B relative to the patient’s specimen. The blue point indicated by the red arrow corresponds to our patient’s specimen. The surrounding tan and orange data points correspond to reference esthesioneuroblastomas of subgroup A and B respectively.

**Table 1 biomedicines-10-02225-t001:** Summary of clinical, biochemical, radiologic, and histologic findings of the reported case. Abbreviations: GH growth hormone, MR magnetic resonance, CS cavernous sinus.

Clinicopathologic Feature	Case Reported
*Patient demographics*	
Age (years)	58
Sex	Female
Race/ethnicity	White, Non-Hispanic
*Patient History*	
Symptoms at presentation	Blurry vision × 1.5 years
Past medical history	Asthma, depression, hypothyroidism
Personal cancer history	None
Family cancer history	Inflammatory breast cancer, lung cancer
*Pituitary involvement*	
Central hypothyroidism	Yes
GH deficiency	Yes
Hypogonadotropic hypogonadism	Yes
Hyperprolactinemia	No
Diabetes insipidus	No
Visual field-testing	Bilateral temporal hemianopsia
*Initial MR imaging findings*	
Lesion location	Sellar and suprasellar
Lesion size (cm)	2.0 × 1.8
Enhancement	Yes
CS invasion	No
*Histopathology—initial resection*	
Histology description	Neurocytic differentiation, abundant neuropil, and fibrillarity
Immunohistochemistry	Diffusely positive for neurofilament and synaptophysin, nuclear reactivity for NeuN; no immunoreactivity for cytokeratin, chromogranin, pituitary hormones, p53, EMA, VIP; infrequent, regional immunoreactivity for GFPA & WT-1
Diagnosis	Ganglioneurocytoma
*Histopathology—repeat resection*	
Histology description	Mixture of small neurocytic cells, larger ganglionic cells, some fibrillary neuropil
Immunohistochemistry	Positive for chromogranin, synaptophysin, neurofilament, and focally for S-100; negative for NeuN, GFPA, PRL, IDH1, p53, reticulin, and cytokeratin CAM 5.2
Diagnosis	Pituitary adenoma with neuronal choristoma (PANCH)

## Data Availability

DNA methylation data are available in GEO: GSE211634.

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
