# Peer review of "DNA Methylation Profiling in Rare Sellar Tumors"

_biomedicines, 2022, doi:10.3390/biomedicines10092225_

Round 1

Reviewer 1 Report

This is an interesting case report that highlights the value of DNA methylation analysis for the correct diagnosis of rare tumors. This is particularly important to guide the treatment.

Nevertheless, given the importance of the DNA methylation analysis for the present study, more  details should be given about the method used, and also about the results obtained. The legend for Figure 2 should explain what are the "dots" presented in different colors, and why "The patient's specimen was most consistent with esthesioneuroblastoma, subgroup A".

Author Response

Dear reviewer, 

Thank you for your response and comments on our manuscript. We sincerely hope that a revised version of the manuscript will still be considered for publication. We have modified the paper in response to your extensive and insightful comments as below:

1. This is an interesting case report that highlights the value of DNA methylation analysis for the correct diagnosis of rare tumors. This is particularly important to guide the treatment.

2. Nevertheless, given the importance of the DNA methylation analysis for the present study, more details should be given about the method used, and also about the results obtained.

Thank you so much for your comment. Details regarding the methods used have been added to section 2.2. Moreover, a more in-depth description of the results generated by the classifier has been added. We hope our methods and results are more clear with these revisions. 

3. The legend for Figure 2 should explain what are the "dots" presented in different colors, and why "The patient's specimen was most consistent with esthesioneuroblastoma, subgroup A".

Thank you so much for your insightful comment. The legend for figure 2 has been modified to include a more in-depth description of the TSNE plot and what each point and color corresponds to. We hope this added explanation improves clarity for the reader. 

We look forward to hearing from you regarding our submission. We would be glad to respond to any further questions and comments that you may have. 

Sincerely,

Nidhi Agrawal, MD

Kyla Wright, BS

Reviewer 2 Report

The manuscript entitled “DNA methylation profiling in rare sellar tumors” is a case report article focused on the usefulness of DNA methylation profiling in diagnosing CNS tumor with atypical clinicopathological and histological features. This study is of importance for biomedical and clinical researchers. However, there some concerns and recommendations.

1.     The authors did not provide a description of the experimental test was used for the identification of DNA methylation in the patient (blood sample or other) presented in this study. This should be given in short in the Abstract and section 2.

2.     The Abstract and Conclusion sections should be re-rewritten because it is not clear what was the finding of this research? i.e. the final diagnosis for the patient based on DNA methylation profiling. Was a diagnosing performed based either on only the methylation profiling or combination with histological and clinicopathological features?

3.     In the Introduction, the authors should provide which DNA methylation signatures and methylation targets associated with brain tumors are known to date and have clinical utility.

4.     In section 2, it is recommended to give a Table that summarizes clinicopathological characteristics of the patient.

5.     There are many inaccuracies in the text. For example, there mistakes should be checked. For example, in the author list, etc. Reference number of GEO should be provided, etc.

6   English language style is not perfect. Some sentences are poorly understood. For example, sentence 4 in the Abstract, and others. Please, check this throughout the all text.

Author Response

Dear reviewer, 

Thank you for your response and comments on our manuscript. We sincerely hope that a revised version of the manuscript will still be considered for publication. We have modified the paper in response to your extensive and insightful comments as below:

The manuscript entitled “DNA methylation profiling in rare sellar tumors” is a case report article focused on the usefulness of DNA methylation profiling in diagnosing CNS tumor with atypical clinicopathological and histological features. This study is of importance for biomedical and clinical researchers. However, there some concerns and recommendations.

1.     The authors did not provide a description of the experimental test was used for the identification of DNA methylation in the patient (blood sample or other) presented in this study. This should be given in short in the Abstract and section

Thank you so much for this important comment. We have added a more detailed description of our methodology, including the sample type used, to both the abstract and section 2.2. We hope you find this explanation useful and succinct.

2.     The Abstract and Conclusion sections should be re-rewritten because it is not clear what was the finding of this research? i.e. the final diagnosis for the patient based on DNA methylation profiling. Was a diagnosing performed based either on only the methylation profiling or combination with histological and clinicopathological features?

Thank you for your comment. The diagnosis was ultimately made using the DNA methylation classifier, as prior phenotypical histopathological analyses were either unrevealing or conflicting. We have added a statement to both our abstract and conclusion that more explicitly states that the results of the DNA methylation classifier resulted in a change in the diagnosis of this patient’s lesion.

3.     In the Introduction, the authors should provide which DNA methylation signatures and methylation targets associated with brain tumors are known to date and have clinical utility.

Thank you for your comment. We agree that the introduction would benefit from an explanation of the clinical utility of DNA methylation profiling in the diagnosis and prognostication of CNS tumors. There are numerous DNA methylation signatures that have clinical utility in neuropathology; thus, we are unable to concisely summarize all previously studied methylation signatures. Instead, we have provided the reader with some examples of how DNA methylation profiling has been demonstrated to more accurately diagnose, further stratify, and predict the prognosis of certain CNS tumors. We hope this gives the reader further context. Thank you.

4.     In section 2, it is recommended to give a Table that summarizes clinicopathological characteristics of the patient.

Thank you so much for this suggestion! We have added a table that summarizes the clinical, biochemical, radiologic, and histopathologic findings of our case.

5.     There are many inaccuracies in the text. For example, there mistakes should be checked. For example, in the author list, etc. Reference number of GEO should be provided, etc.

Thank you so much for pointing out our error in the author list; this has been corrected. Unfortunately, our GEO reference number is still pending at this time. After discussion with the manuscript’s assigned editor, we thought it was best to submit the revisions at this time to ensure a timely review process. We are actively working on uploading the DNA methylation data to GEO, and will ensure a GEO reference number is added to the manuscript before publication if accepted. Thank you so much and we hope this does not cause any inconvenience during your review.

6   English language style is not perfect. Some sentences are poorly understood. For example, sentence 4 in the Abstract, and others. Please, check this throughout the all text.

Thank you so much for your comment. The manuscript has been proofread and checked for grammar errors by multiple authors that are native English speakers. Some shorthand was used at points within our original text, and we have made changes to improve clarity. I hope you find the text is more easily understood with our edits and we would be happy to make additional changes if needed. Thank you!

We look forward to hearing from you regarding our submission. We would be glad to respond to any further questions and comments that you may have. 

Sincerely,

Nidhi Agrawal, MD

Kyla Wright, BS

Round 2

Reviewer 2 Report

The authors have properly addressed all my concerns and recommendations.